# OpenReview forum: "MobileGuard: Safeguarding Mobile Task Automation"
_ICLR.cc/2026/Conference — ICLR 2026 Conference Withdrawn Submission_

### Official Review · Reviewer_D1ua · 2025-10-26

**Soundness:** 3
**Presentation:** 3
**Contribution:** 2
**Rating:** 6
**Confidence:** 3

**Summary:**

The paper addresses an important and underexplored problem—safety evaluation for mobile automation agents—and contributes a valuable dataset and platform.

**Strengths:**

1.	The work pioneers the explicit safety evaluation of mobile agents, which is paramount for real-world deployment. By shifting the focus from mere action accuracy to the consequences of actions on user data, the paper establishes a necessary new line of research..
2.	The paper provides a robust and valuable conceptual contribution through the formalization of unsafe transitions categorized as Irreversible Loss (IL), Unintended Modification (UM), and External Broadcast (EB).
3.	The paper provides powerful, systematic evidence demonstrating that current mobile agents consistently exhibit critical safety blind spots.
4.	The writing is clear, with figures and tables well presented to support the claims.

**Weaknesses:**

1.	The inter-annotator agreement (75.8%) suggests that the labeling of unsafe transitions is somewhat subjective. More details about the annotation process, conflict resolution, and label uncertainty would strengthen the dataset’s credibility.
2.	The classification of unsafe transitions is reasonable, but the risk level of certain operations may depend on user intent and contextual factors, which static annotations cannot fully capture.
3.	The benchmark includes only seven Android apps and 1,953 actions, which limits coverage of the broader mobile ecosystem. Additional app categories (e.g., financial, accessibility) or cross-platform evaluations would improve generalization and strengthen empirical conclusions.

**Questions:**

Refer to the weaknesses.

---

### Official Review · Reviewer_1u2g · 2025-10-30

**Soundness:** 3
**Presentation:** 3
**Contribution:** 3
**Rating:** 4
**Confidence:** 3

**Summary:**

MobileGuard is a benchmark designed to evaluate mobile automation safety, using the concept of unsafe-transitions. It categorizes three error types. (1) irreversible loss of user data such as deleting a playlist or photo (2) unintended modification of the mobile application state such as changing privacy settings (3) external broadcast to application servers such as submitting payments or sharing content without user permission. One key difference from existing work is that MobileGuard focuses on scenarios when the agent is autonomously exploring the environment; therefore, task information is not provided, and the safety-transitions

**Strengths:**

* The paper introduces a new axis of agentic safety: the safety performance of autonomous agents in mobile environments.
* The evaluation results indicate that agents fail to identify unsafe transitions, meaning that this feature is currently underexplored when developing mobile agents. However, this relies on the soundness of the benchmark, for which I have some related questions below. If this is sufficiently answered, I am willing to raise my score.

**Weaknesses:**

* One additional category of models to consider is external guardrail models (e.g., WildGuard, LlamaGuard). It would be interesting to see if recent external guardrail models fail to detect unsafe transitions.
* The inter-annotator agreement rate is reported to be 75.8% in the introduction section, but there is no mention of this result in the main sections & appendices. How this agreement rate was obtained should be included in the main sections or appendices.
* In Figure 5 of Appendix A, pressing the backspace on a number 1 in the calling app seems to be considered an unsafe transition. Although this action does remove some information, just deleting a single number '1' does not seem to be critically related to unsafe transitions. In regard to this, please see my first question.
* (Minor) Mobile-Agent-v2 seems to be similar to MobileGPT, and UI-Tars with AutoDroid-v2. It might be better to present Table 2 together or replace Table 3 if there is no clear reason to divide the results. It might be misleading to divide the tables and present weaker agents as the main results in Table 2.

**Questions:**

* Please provide additional information on how these tasks were verified to represent mobile safety. Specifically, who verified the tasks, and how did they verify that these tasks are safety-relevant?
* How was the inter-annotator agreement rate obtained?

---

### Official Review · Reviewer_aihL · 2025-10-30

**Soundness:** 2
**Presentation:** 3
**Contribution:** 2
**Rating:** 4
**Confidence:** 4

**Summary:**

This paper introduces MobileGuard, a benchmark dedicated to evaluating safety in mobile task automation. The authors formalize the concept of unsafe transitions, categorized into irreversible loss, unintended modification, and external broadcast, and construct a dataset containing labeled unsafe transitions across seven popular mobile applications. The benchmark is built using a human-in-the-loop annotation process combining LLM-assisted labeling and manual verification. Evaluations of multiple state-of-the-art agents (e.g., MobileGPT, MM-Navigator, AutoDroid-v2) reveal significant weaknesses in identifying unsafe actions. The study also explores mitigation strategies such as few-shot prompting and fine-tuning, showing limited but measurable improvements.

**Strengths:**

- The MobileGuard dataset is carefully curated through a multi-stage annotation process, combining automated LLM preprocessing and rigorous human verification. It includes HTML representations, screenshots, and XML hierarchies, enabling rich and reproducible evaluation.
​​- The evaluation​​ is comprehensive: The study evaluates a diverse set of agents, including LLM-based, VLM-based, and SLM-based, under both zero-shot and few-shot Chain-of-Thought settings. The results consistently highlight safety limitations across models, offering valuable empirical insights.

**Weaknesses:**

- The benchmark assesses action risks independently of user tasks, which is conceptually problematic. For example, actions like "subscription" or "deletion" may be intentional in certain tasks. While the authors position the benchmark for "safe exploration," the experiments do not evaluate exploration systems, limiting practical relevance.
- ​​Although an "extensible emulator-based evaluation platform" is mentioned, no technical details or experiments are provided about this. This omission hinders reproducibility and validation of the platform’s utility.
- The dataset covers only seven apps, which may not capture the full spectrum of mobile interaction risks. Moreover, evaluations are conducted statically rather than in dynamic task environments, reducing real-world applicability.
- The paper claims to be the "first benchmark for mobile automation safety," but fails to cite relevant works such as AgentHazard (https://arxiv.org/abs/2507.04227), which also addresses security vulnerabilities in mobile GUI agents. This weakens the novelty claim.
- Only a limited set of agents are tested, and recent models are only briefly mentioned. Broader inclusion of emerging agents would strengthen generalizability.

**Questions:**

See weaknesses. I'm especially interested in the details of emulator-based evaluation platform.

**Details Of Ethics Concerns:**

The annotation process involves human efforts.

---

### Official Review · Reviewer_H4zb · 2025-10-31

**Soundness:** 2
**Presentation:** 2
**Contribution:** 3
**Rating:** 2
**Confidence:** 4

**Summary:**

This paper introduces MobileGuard, the first benchmark specifically designed to evaluate safety in mobile automation. It formalizes UI-grounded unsafe transitions, which are categorized as irreversible loss, unintended modification, and external broadcast. It provides a dataset covering 7 apps, 1,953 actions, and 269 unsafe cases, along with an evaluation pipeline. Experiments show that current state-of-the-art agents consistently fail to detect unsafe actions across diverse mobile apps, while few-shot CoT prompting and fine-tuning yield moderate improvements, they remain insufficient for ensuring safety. The paper further classifies failures into three major error types and positions MobileGuard as a foundation for future research toward safer mobile agents.

**Strengths:**

1.	Introduces the first formal definition of mobile automation safety through the concept of unsafe transitions, which is grounded in established HCI usability principles.
2.	Builds MobileGuard, the first benchmark and emulator framework for evaluating safety in mobile task automation, featuring real-world data from seven apps and detailed unsafe-action annotations verified by humans.
3.	Provides systematic evaluation across multiple LLM-, VLM-, and SLM-based agents, revealing consistent safety failures and classifying them into three interpretable error types, offering diagnostic insights for future research.

**Weaknesses:**

1.	The dataset currently includes only 7 apps and 269 unsafe cases, which limits the diversity of UI layouts, interaction patterns, and application domains represented. Broader coverage would help validate the generalizability of the benchmark.
2.	The reported inter-annotator agreement (75.8%) suggests some ambiguity in interpreting unsafe transitions. A more detailed discussion or validation of borderline cases could strengthen confidence in labeling consistency.
3.	The LLM-assisted annotation pipeline is efficient but not deeply analyzed, and potential systematic biases introduced by the model remain unexplored.
4.	The work primarily diagnoses safety issues rather than proposing mechanisms to mitigate or model them. Providing design insights or preliminary directions for safety-aware agent training could enhance its long-term impact.

**Questions:**

1.	How do the authors ensure the consistency of the unsafe transition definition across different apps and task contexts, given that action semantics can vary widely between interfaces?
2.	Are the three risk categories (IL, UM, and EB) mutually exclusive by design, or can overlaps occur? Are there potential unsafe cases not covered by these categories?
3.	To what extent does the notion of “unsafe” depend on user intention or contextual goals? Were different task scenarios considered during annotation to account for context-dependent safety?
4.	Could the authors provide more details about the LLM-assisted annotation prompts and the human verification process? Were safe transitions also manually reviewed to ensure labeling balance and reliability?
5.	Do the authors plan to release the dataset, and evaluation pipeline to facilitate reproducibility and future research on mobile agent safety?

---

### Note · Authors · 2025-11-14

I have read and agree with the venue's withdrawal policy on behalf of myself and my co-authors.